# Naples Prognostic Score Predicts Tumor Regression Grade in Resectable Gastric Cancer Treated with Preoperative Chemotherapy

**DOI:** 10.3390/cancers13184676

**Published:** 2021-09-17

**Authors:** Eva Lieto, Annamaria Auricchio, Giuseppe Tirino, Luca Pompella, Iacopo Panarese, Giovanni Del Sorbo, Francesca Ferraraccio, Ferdinando De Vita, Gennaro Galizia, Francesca Cardella

**Affiliations:** 1Division of GI Tract Surgical Oncology, Department of Translational Medical Sciences, Vanvitelli University, 80132 Napoli, Italy; annamaria.auricchio@unicampania.it (A.A.); giovanni.ds88@gmail.com (G.D.S.); gennaro.galizia@unicampania.it (G.G.); francescacardella@gmail.com (F.C.); 2Division of Medical Oncology, Department of Precision Medicine, Vanvitelli University, 80132 Napoli, Italy; giuseppe.tirino@unicampania.it (G.T.); luca.pompella@unicampania.it (L.P.); ferdinando.devita@unicampania.it (F.D.V.); 3Division of Pathology, Department of Mental and Physical Health and Rehabilitation Medicine, Vanvitelli University, 80132 Napoli, Italy; iacopo.panarese@unicampania.it (I.P.); francesca.ferraraccio@unicampania.it (F.F.)

**Keywords:** gastric cancer, tumor regression grade, neoadjuvant treatment, Naples prognostic score

## Abstract

**Simple Summary:**

Multimodal treatment of locally advanced gastric cancer is still debated today due to controversial results in different trials. Nevertheless, perioperative chemotherapy with radical surgery certainly shows a better long-term outcome than surgery alone, so much so it is the main multimodal treatment offered in Europe, at the present. Tumor regression grade is the objective response to preoperative chemotherapy and its extent, in terms of reduction of neoplastic cells in the resected specimen, is strongly affected by Lauren’s classification, TNM stage, and tumor grading. Therefore, since this information can be achieved only after surgical resection, the return of chemotherapy is quite unpredictable in advance and, in about half cases, it is definitely ineffective. Naples Prognostic Score, that mirrors the immune–nutritional conditions, tested on 59 consecutive advanced gastric cancer patients undergoing multimodal treatment, showed a strong power in predicting tumor regression grade and therefore is strictly correlated with long-term outcome and survival.

**Abstract:**

Despite recent progresses, locally advanced gastric cancer remains a daunting challenge to embrace. Perioperative chemotherapy and D2-gastrectomy depict multimodal treatment of gastric cancer in Europe, shows better results than curative surgery alone in terms of downstaging, micrometastases elimination, and improved long-term survival. Unfortunately, preoperative chemotherapy is useless in about 50% of cases of non-responder patients, in which no effect is registered. Tumor regression grade (TRG) is directly related to chemotherapy effectiveness, but its understanding is achieved only after surgical operation; accordingly, preoperative chemotherapy is given indiscriminately. Conversely, Naples Prognostic Score (NPS), related to patient immune-nutritional status and easily obtained before taking any therapeutic decision, appeared an independent prognostic variable of TRG. NPS was calculated in 59 consecutive surgically treated gastric cancer patients after neoadjuvant FLOT4-based chemotherapy. 42.2% of positive responses were observed: all normal NPS and half mild/moderate NPS showed significant responses to chemotherapy with TRG 1–3; while only 20% of the worst NPS showed some related benefits. Evaluation of NPS in gastric cancer patients undergoing multimodal treatment may be useful both in selecting patients who will benefit from preoperative chemotherapy and for changing immune-nutritional conditions in order to improve patient’s reaction against the tumor.

## 1. Introduction

Gastric cancer (GC) is still the third most common cause of cancer-related death worldwide, despite the huge advancements in its diagnostic and therapeutic management we can observe today [1].

Although radical resection with adequate lymphadenectomy (so called R0 surgery with D2 gastrectomy) is the first-line treatment in non-metastatic setting, chemotherapy and radiotherapy both represent an essential therapeutic approach in advanced GC stages [2].

Since 2000s, a multimodal approach to locally advanced or upfront resectable GC has become the new gold standard, as several trials have shown the efficacy of a combined treatment compared with surgery alone [3,4,5]. Nevertheless, a worldwide standard of treatment does not exist yet, owing to controversial results that are reported in different trials [6,7,8,9,10].

Perioperative chemotherapy with FLOT-4 [11] is currently a step of multimodal treatment of advanced GC in Europe, with proven benefits on the likelihood of radical resection and 5-year survival. In particular, the effects of preoperative chemotherapy on tumor downstaging are predictive of a better outcome, both because a less demolitive surgery could be accomplished and possible microscopic remaining tumoral tissue eliminated [12].

Unfortunately, in about 50% of cases this kind of therapy is ineffective [13]. TNM stage, differentiation, and Lauren’s classification [14] of the tumor are all independent prognostic factors affecting GC outcome and are strictly related to chemotherapy efficacy, particularly in preoperative setting [15,16].

Tumor regression grade (TRG) is the measurable histological effect of preoperatively given chemotherapy, and predictive of long-term advanced GC outcome [17]. A TRG1–3 is strongly related with a better GC survival than TRG 4–5 [18]. Therefore, predictivity of TRG extent would be decisive in planning multimodal treatment; knowing in advance the likelihood of chemotherapy effectiveness would allow a motivated therapeutic choice about different drugs to be used and if it is worth doing it in that patient. Moreover, it is well known that different patients in the same tumor stage can have different responses to the same therapy, as other factors might affect the cancer treatment outcome.

A properly selection of patients in term of clinical staging is crucial for the identification of advanced GC patients who will benefit from perioperative therapy, with a proven effective TRG. Since the exact TNM staging is not completely achieved by current radiological imaging [19,20], and a relative predictable effectiveness of preoperative chemotherapy is quite impossible until TRG is obtained, we have investigated whether other prognostic factors, such as immune–nutritional patient status could affect clinical and pathological response to preoperative chemotherapy in advanced GC patients [21].

Additionally, tumor growth is also strongly affected by the host response, in terms of systemic inflammation and nutritional status [21,22,23]. Patient response to the tumor may be crucial, and the investigation in this field is one of major challenge of the last years. Several score systems exist today that investigate on host environment, with a strong prognostic power when tested in many malignant tumors [24,25,26,27,28,29,30,31].

We have previously published the Naples Prognostic Score (NPS) and tested its validity both in colorectal and gastric cancer patients undergoing radical surgical resection [32,33]. NPS is easily calculated by combining Neutrophil to Lymphocyte Ratio (NLR), Lymphocyte to Monocyte Ratio (LMR), Albumin and Cholesterol serum levels. This score perfectly matches immune and nutritional status of the patient and is an independent prognostic factor of surgical outcome and distant survival.

In the present study we have retrospectively tested NPS in predicting the effectiveness of neoadjuvant chemotherapy regimen in terms of TRG, and long-term outcome in GC patients undergoing radical surgery after neoadjuvant FLOT4-based chemotherapy regimen.

Knowing in advance tumor sensitivity to preoperative chemotherapy may become crucial, in the future, for assessing the best therapeutic planning in a patient whose predictable response could be the guide in multimodal approach.

## 2. Materials and Methods

From January 2017 to June 2021, 98 consecutive patients with non-metastatic biopsy-proven gastric adenocarcinoma were admitted to Divisions of Medical and Surgical Oncology of Vanvitelli University, (Napoli, Italy). All patients had careful evaluation including clinical examination, laboratory tests, abdominal echography, computed tomography (CT) scans of the neck, chest, and abdomen, and endoscopic ultrasonography (EUS). Particularly, pretreatment or basal cancer staging was performed by comparing evaluation of tumor parietal growth (T parameter), and lymph node status (N parameter) by means of CT scan and EUS. Patients with gastric adenocarcinoma judged to invade the muscularis propria or beyond (≥T2) and/or to be node positive (cN+) were initially considered for multimodal treatment including perioperative chemotherapy and surgery. Six patients with node-negative early gastric cancer (T1sm) were excluded and underwent up-front surgery. Out of remaining 92 patients, 10 with severe gastric bleeding requiring several blood transfusions, fifteen with gastric stenosis and severe nutritional impairment, and six not-fit for chemotherapy, underwent up-front surgery. Overall, 61 patients remained eligible and were treated with multimodal treatments.

### 2.1. Multimodal Treatment

In total, 61 patients were scheduled to receive four preoperative and four postoperative 2 week cycles of 50 mg/m^2^ docetaxel, 85 mg/m^2^ oxaliplatin, 200 mg/m^2^ leucovorin on day 1, and 2600 mg/m^2^ fluorouracil as 24 h infusion on day 1 (FLOT-4 regimen) [11]. Toxic effects were assessed before starting and at each 2 week cycle using the Common Terminology Criteria for Adverse Events (version 5.0) published on 27 November 2017 by the U.S. Department of Health and Human Services [34]. Treatment deferral and dose modification were based on the results of a complete hematologic evaluation performed on the day of the planned treatment. When grade >2 thrombocytopenia or neutropenia or other significant nonhematologic toxic effects occurred, chemotherapy was delayed for up to 2 weeks. Fluorouracil dose was reduced in cases of grade >3 diarrhea, stomatitis, and dermatitis. Peripheral sensitive neuropathy was graded according to the following oxaliplatin-specific scale: grade 1, paresthesia or hypoesthesia of short duration with complete recovery before the next cycle; grade 2, paresthesia or hypoesthesia persisting between two cycles without functional impairment; and grade 3, permanent paresthesia or hypoesthesia resulting in functional impairment. Oxaliplatin dose was reduced for grade 3/4 neutropenia or thrombocytopenia and in cases of persistent (>14 days) paresthesia or temporary (7–14 days) painful paresthesia or functional impairment. In cases of persistent (>14 days) painful paresthesia or functional impairment, oxaliplatin was omitted from the treatment until recovery. Nutritional counseling was provided in all cases. One patient experienced worsening of health condition and did not undergo further treatment. Due to hematological toxicity, 5 of the 60 patients (8%) were only able to practice three cycles of chemotherapy. Complete re-staging, including CT scan and EUS, and surgery were scheduled for 3 and 4 weeks after the last dose of preoperative chemotherapy, respectively. Disease progression with hepatic and pulmonary metastases were discovered in one patient and an unsuccessful second line chemotherapy was performed.

Overall, 59 patients underwent surgery. Macroscopic type was assessed by Borrmann classification [35]. Lauren’s classification was used to define the histological type dividing intestinal from diffuse/mixed tumors [36,37]. The histological grade was established by dividing well-moderately differentiated tumors from poorly differentiated and signet ring cells [38]. Tumor regression grade was assessed by using the system described by Mandard in 1994 [17] and was carefully checked by two independent pathologists blinded each other and with no prior knowledge of clinicopathological parameters. Discrepancies between investigators (<5% of the cases) required a third joint observation with conclusive agreement. The system defines five grades of tumor response [17,39]. Complete disappearance of the tumor corresponded to grade 1; fibrosis with scattered tumor cells was grade 2, and fibrosis and tumor cells with preponderance of fibrosis was grade 3, respectively. These grades were judged as total or partial tumor regression. On the contrary, fibrosis and tumor cells with preponderance of tumor cells (grade 4), and tissue of tumors without changes of regression (grade 5), were judged as scarce or no response to the chemotherapy [40,41].

Finally, all surgical operations were successful with no postoperative death. Four (7%) mild to severe postoperative complications (two abdominal bleeding and anastomotic dehiscences) were conservatively treated. All patients recovered and were discharged; 42 (71%) received postoperative chemotherapy (≥1 cycle).

### 2.2. Naples Prognostic Score

According to our previous experiences in colorectal and gastric cancers, NPS was calculated based on the following four parameters: serum albumin (normal: ≥4 g/dL), total cholesterol (normal: >180 mg/dL), LMR (normal: ≤2.96), and NLR (normal: >4.44) [32,33]. Each parameter was assigned a score (normal value = 0, altered value = 1), and patients were initially assigned a score of 0 to 4 (NPS score). Afterwards, patients were divided into three groups (NPS group): patients with score = 0 were assigned to group 0 (normal values for all 4 parameters); patients with score 1 or 2 were assigned to group 1 (one or two altered values); and patients with score 3 or 4 were assigned to group 2 (three or four altered values).

### 2.3. Statistical Analysis

Continuous variables are expressed as range, mean ± standard deviation, and median. Some variables (age and tumor size) were dichotomized by using median values, or normal values (carcinoembryonic antigen or CEA, and Carbohydrate Antigen or Ca19-9). The chi-square test was used for modeling the relationship between TRG and other prognostic factors. Multivariate analysis with multiple logistic regression was used to individuate independent prognostic variables related to TRG. Multicollinearity among variables supposed to have high correlation was investigated with interaction analysis. The receiver operating characteristic (ROC) curve was applied to quantify the performance of different factors to predict neoadjuvant chemotherapy effects by computing the area under the curve (AUC), sensitivity, specificity, positive predictive (+PV), and negative predictive (−PV) values. Particularly, +PV indicates the probability of therapeutic failure when the specified factor is present; on the contrary, −PV indicates the probability of a good therapeutic effect when the variable is absent. After that, a comparison ROC curve among significantly independent factors was depicted. The Kaplan–Meier method and long-rank test were used to compare survival curves by providing hazard rate (HR) with its 95% confidence interval (CI) and *p* value. All analyses were two-sided, and *p* < 0.05 was considered to be statistically significant. Statistical analyses were carried out using the SPSS 21.0 software (SPSS Inc., Chicago, IL, USA), integrated by the Medcalc^®^ software version 12.5.0.0 (Mariakerke, Belgium).

## 3. Results

### 3.1. Clinical Characteristics

Age ranged from 31 to 83 years; mean 61 ± 11; median 63, with 41 male and 18 female. CEA serum levels ranged from 0.4 to 367 ng/mL; mean 10.2 ± 47, median 2.7. Ca19-9 serum levels ranged from 0.6 to 4553; mean 133 ± 605, median 13. Almost 40% of tumors invaded the upper third of the stomach, and in eight cases (13%) the cancer occupied the entire organ. Nearly 60% of cases were types 1 and 2 of the Bormann classification. Tumor size ranged from 0.5 to 15 cm; mean 4 ± 3, median 3. Nearly two-thirds of tumor presented an undifferentiated grade, and over half (34 cases—57%) showed a diffuse type. Preoperative staging with endoscopic ultrasonography and abdominal CT scan showed an advanced gastric cancer in nearly 90% of the cases with only two and five patients presenting a T1 or T2 tumor, respectively. Presence of metastatic nodes were suspected in 34 cases (57%).

### 3.2. Tumor Regression Grade and Its Prognostic Predictors

After the operation, in three (5%) patients pathological findings showed complete tumor regression (Mandard grade 1). Grade 2 and grade 3 were observed in nine (15%) and 13 (22%) of patients, respectively. In the remaining 34 patients, 15 (25%) showed grade 4, and 19 (32%) showed grade 5, respectively. Thus, total or partial tumor regression was observed in 42.4% of the cases (25 patients) [40,41]. Failure to respond to neoadjuvant chemotherapy was significantly related to distal or diffuse tumor site, advanced tumor infiltration, node metastasis, advanced Bormann and diffuse Lauren types, poor differentiated tumor, and large tumor size (Table 1). In addition, pretreatment NPS evaluation showed to be strongly correlated with TRG. Both NPS score and NPS group appeared to be able to significantly predict the response to the therapy. All patients with normal NPS (score = 0) and half of the patients, showing mild to moderate NPS values (NPS score < 2), positively responded to neoadjuvant treatment. On the contrary, only 20% of patients with the worst NPS values (NPS scores 3 or 4) appeared to benefit from preoperative treatments. However, at multivariate analysis with multiple logistic regression, where interaction analysis excluded any correlation, some factors lost their significance (Table 2). Tumor regression grade was related only to Lauren type, histological grade, and NPS. In addition, also tumor depth (T parameter) and tumor size showed a slight significant trend.

The Receiver Operating Characteristic (ROC) curves were analyzed for each factor known to correlate with TRG (Table 3). In total, 83, 75, and 77% of patients with diffuse type, or poorly differentiated tumor, or NPS score 3/4 (NPS group = 2) showed a neoadjuvant chemotherapy failure, respectively (Figure 1). In addition, 69, 79, and 71% of patients with intestinal type, or differentiated tumor, or NPS score 0–2 (NPS group = 0–1) showed a good tumor regression grade, respectively. Interestingly, among 11 patients showing, at the same time, intestinal type and differentiated tumors with NPS score ranging from 0 to 2, all but one presented a favorable TRG (3 TRG = 1, 5 TRG = 2, and 2 TRG = 3), respectively. Only one patient had a TRG = 4. On the contrary, among 13 patients with unfavorable characteristics (diffuse and poor differentiated tumors with NPS score 3/4) all showed a disappointing tumor regression grade (in four patients, TRG was = 4, and in nine patients was = 5), respectively.

### 3.3. Tumor Regression Grade, Surgical Radicality and Survival Rate

Although the aim of this work was the analysis of the factors related to TRG, it seemed appropriate to also analyze the relationship between TRG with surgical radicality and survival rate. There were 46 radical operations (78%); in the remaining 13 cases, eight patients underwent a microscopic or macroscopic tumor residual (so called R1/R2 gastric resection) and five patients a surgical bypass or explorative laparotomy. TRG was strongly related to surgical radicality; among 46 resectable patients, 24 had a favorable TRG, and 22 showed a poor response. On the contrary, among 13 patients with non-radically resectable tumors, only one showed a response to preoperative treatment (*p* = 0.0108). The follow-up time ranged from 2 to 52 months, mean 18 ± 13, median 17. At the end of study, 33 (56%) patients are alive, and 26 are dead. Among 46 patients undergoing radical surgery, 27 (59%) are alive and 17 are died. According to the tumor regression grade, the 1 to 4 yrs overall survival rates (Figure 2A) were 91, 74, 74, and 55% in patients showing favorable tumor regression grade, and 69, 38, 18, and 9% in patients with little to no response to preoperative chemotherapy, respectively (HR = 0.26, 95%CI 0.12–0.57, *p* = 0.0021).

In radically resected patients (Figure 2B), the 1 to 4 years survival rates were 91, 79, 79, and 59% in patients showing favorable tumor regression grade, and 85, 45, 18, and 18% in patients with little to no response to preoperative chemotherapy, respectively (HR = 0.27, 95%CI 0.10–0.71, *p* = 0.0078). Overall, the hazard ratio was 0.26 so that the estimated relative risk of oncological death in patients with a grade 1–3 tumor regression was 26% of that in non-responder patients. In radically resected patients a favorable tumor regression grade reduced the risk of death at 4 years by about three quarters (HR = 0.27). In addition, in Figure 3, the overall survival rate according to the NPS score is shown, confirming the previous report on its prognostic significance in gastric cancer patients undergoing surgery [33].

## 4. Discussion

This study confirms previous suggestions that in gastric cancer patients, treated with neoadjuvant chemotherapy, tumor regression grade is significantly correlated with overall survival rate. Importantly, in naïve patients TRG can be predicted with sufficient determination by Lauren’s classification, histological grade, and Naples prognostic score, a novel scoring system that mirrors inflammatory and immune-nutritional status of the patient, and has already been demonstrated to be a powerful indicator of outcome in several human tumors.

Prognosis of advanced GC is still poor today, since its high recurrence rate and a long-term survival rate of less than 30%, mostly in Western Countries [42]. Unfortunately, current lack of a screening program in adult population does not allow an early diagnosis in which R0 surgery alone could be effective. Multimodal treatment of advanced gastric cancer, whose efficacy is widely recognized, is conversely controversial until now, because of different results in the most accredited trials [6,7,8,20]. Therefore, a standard of treatment is not acknowledged worldwide yet and multimodal therapeutic approach is different in different Countries. Since 2019, FLOT4 perioperative chemotherapy is the standard of care in most of European Centers for operable locally advanced GC, showing a greater effectiveness in terms of overall survival compared with perioperative therapy ECF (epirubicin, cisplatin, fluorouracil) or ECX (epirubicin, cisplatin, capecitabine) -based regimens [10,11,43,44].

TNM tumor stage, histological grade, and Lauren’s classification strongly affect long-term GC outcome and are important prognostic factors also in predicting FLOT-based perioperative chemotherapy responsiveness [11].

Downstaging of the primary tumor, which increases resectability rates and the possibility of achieving a complete (R0) resection, as well as elimination of occult micro-metastatic disease, constitute the main rationale for adding chemotherapy whenever feasible. Tumor regression grade, that reflects such effects, has emerged as an important prognostic factor in several human tumors, including gastric cancers [45,46]. For gastrointestinal malignancies, some classifications have been proposed, particularly by Mandard [17] and Becker [47], and it has been shown that the prognostic value of TRG may even exceed those of currently used staging systems (e.g., TNM staging) for tumors treated by neoadjuvant therapy [39].

In this study we used the Mandard classification that is easy and understandable. Although previous experiences had evidenced that only 1–2 Mandard grades correlated with better outcome in gastric cancers undergoing neoadjuvant treatment, more recent data have elucidated that the first three Mandard grades are strictly related to long-term survival rates [39,40,41]. For this reason, we decided to consider together total and partial tumor regression (grade 1 and 2–3, respectively) and compare them with scarce or no response (grades 4 and 5, respectively). In addition, since the most critical issue regarding this topic is inter- and intra-observer variability and the lack of standardization, a strict rule of diagnosis with two skillful pathologists who separately analyzed each specimen was adopted; reserving to a joint analysis the very few cases of discrepancies [48].

Unfortunately, we cannot know TGR until resection is accomplished after preoperative chemotherapy; so, we cannot assess whether FLOT4 regimen will be effective in that patient. For this reason, perioperative chemotherapy is indiscriminately recommended in all patients with gastric cancer suspected to be T2 or more, and/or node positive. Therefore, it would be crucial to predict the effect of the treatment in patient management because, among ineffective cases, perioperative chemotherapy will be useless and delay a curative intervention. To date, diffuse Lauren’s type and poor differentiation of the tumor, which are easily obtained by endoscopic biopsy, have been considered the most important predictors of chemotherapy failure; unfortunately, several contrasting results about these features do not contraindicate FLOT4 in such patients [43].

There is growing evidence that not only tumor status but also host characteristics, particularly inflammatory and immune-nutritional status, are crucial for tumor growth and cancer progression influencing the course of several human malignancies, including gastric cancer [49]. We have recently demonstrated that NPS is a powerful predictor of long-term survival both in colorectal- and gastric-cancer patients [32,33]. Subsequently, many independent studies from different parts of the world, have shown that NPS is significantly correlated with oncological outcome in several human tumors including lung, endometrial, pancreatic, and gastro-esophageal cancers [28,50,51,52,53]. This is not surprising because NPS includes the neutrophil-to-lymphocyte and the lymphocyte-to-monocyte ratios which have been shown to be reliable surrogate indicators of the host inflammatory and immunological status [54,55]. In addition, serum levels of albumin and cholesterol have been demonstrated to correctly reflect the nutritional status but also mirror systemic inflammation since their concentrations may be reduced by pro-inflammatory substances, such as cytokines [56].

In this study, we tested NPS in 59 GC patients, perioperatively treated with FLOT4 regimen, and the percentage of total or partial tumor response was 42.4%. From the implementation, in the last years, of different tumor regression grading systems, the percentages of tumor response to preoperative chemotherapy ranged from 20 to 60% according to a large variability of definitions, analyses, tumor characteristics, classifications, and chemotherapy regimens [41]. However, the available data with FLOT4 indicate the total or partial response rate is about 40%, which is comparable with the current series [43].

TRG showed to be strictly related to NPS. All with normal, and half of the patients with moderate NPS score responded to neoadjuvant therapy, while less than 20% of patients with worse NPS benefited from the treatment. In addition, an effective tumor response was significantly observed in intestinal type and well-moderately differentiated tumors. In contrast, diffuse type and undifferentiated and/or signet ring cells tumors showed scarce or absent response. Together NPS, histological grade and Lauren’s classification confirmed their relations with TRG. These findings may have important clinical implications. As two latter prognostic factors of TRG are not modifiable, NPS instead could be ameliorated by improving malnutrition and by using anti-inflammatory agents, such as aspirin or other nonsteroidal anti-inflammatory drugs. Indeed, these drugs have been shown to attenuate systemic inflammation and cachexia, thus ameliorating tolerance of anticancer therapy and long-term outcome [31,32,33]. In addition, patients showing negative predictive factors, such as diffuse type, undifferentiated tumors, and high NPS grade should be scheduled for a different or enhanced neoadjuvant chemotherapy, including biologic agents and immunotherapy.

This study presents some points and limitations which deserve discussion. Firstly, retrospective nature and smallness of the sample. Nevertheless, the analysis has been carried out on a strictly homogeneous group of GC patients; in fact, despite their small number, almost all patients could receive the same preoperative chemotherapeutic regimen, and only two of them were excluded for hematological toxicity and health conditions worsening. On the other hand, all patients were surgically treated by the same team; as a result, uniformity in technical modalities has been widely observed. Finally, the short study length ensures further homogeneity that allows the above limits to be partially overcome. Secondly, the 4 year survival rates could change with longer follow-up time as many patients were censored at earlier time points. However, it is similar to most studies, and in patients with advanced gastric cancer, this time of follow-up appears sufficiently adequate and provides robust statistical analysis [11].

## 5. Conclusions

In conclusion, our series NPS is a strong predictor of effectiveness of preoperative FLOT4 regimen and it can be successfully used to select patients who more likely will benefit from a neoadjuvant treatment.

## Figures and Tables

**Figure 1 cancers-13-04676-f001:**
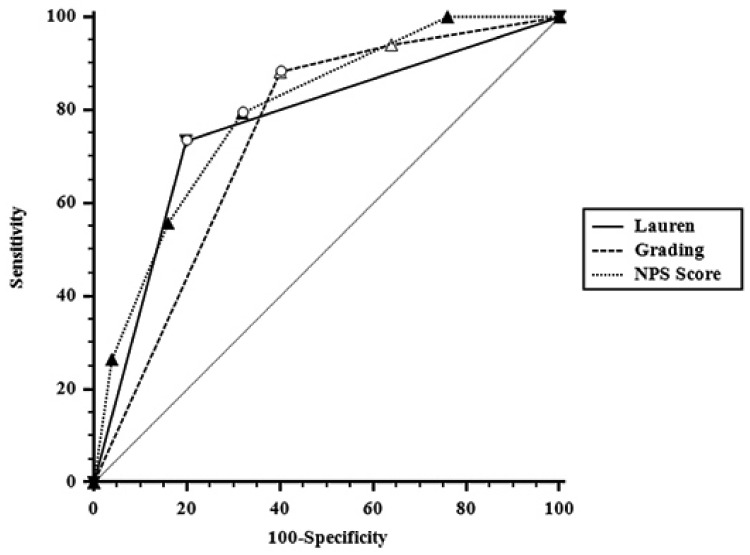
Comparison of three receiver operating characteristics (ROC) curves referred to Lauren’s classification, tumor grade, and NPS score towards tumor grade regression. The analysis offers a complete sensitivity/specificity report and a graph that plots the true positive rate in function of the false positive rate at different cut-off points. The line for each factor is far away from the diagonal, and there were no differences in the area under the curve among the three factors (Lauren vs. tumor grade *p* = 0.7881; Lauren vs. NPS score *p* = 0.7096; tumor grade vs. NPS score *p* = 0.5535).

**Figure 2 cancers-13-04676-f002:**
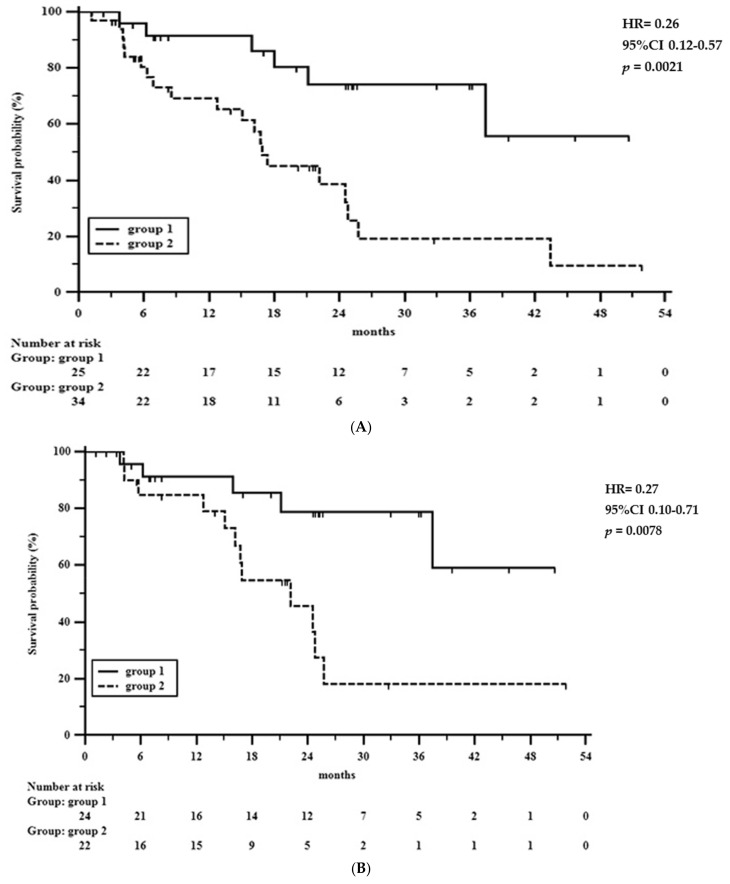
(**A**) 1 to 4 year overall survival rate in 25 patients showing grade 1–3 tumor regression (group 1) and in 34 patients with grade 4–5 tumor regression (group 2). (**B**) The 1 to 4 year overall survival rate in patients undergoing radical surgery; 24 patients showing grade 1–3 tumor regression (group 1) and in 22 patients with grade 4–5 tumor regression (group 2).

**Figure 3 cancers-13-04676-f003:**
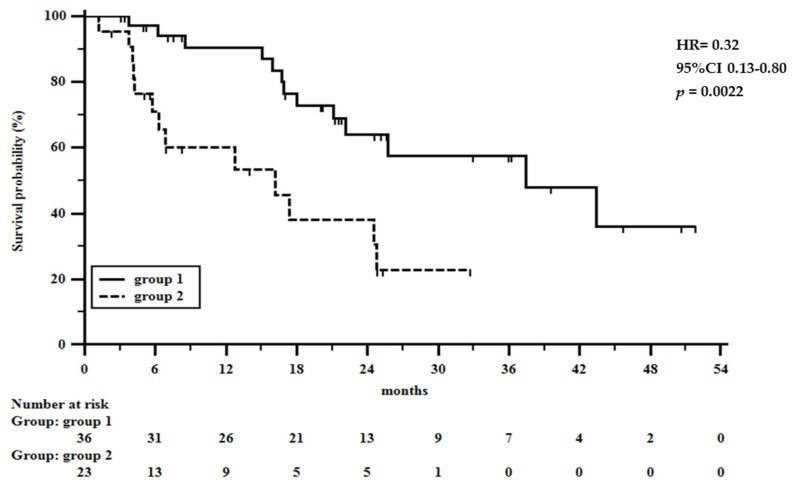
The 1 to 4 year overall survival rate in 36 patients showing NPS score 0–2 (group 1) and in 23 patients with NPS score 3 or 4 (group 2).

**Table 1 cancers-13-04676-t001:** Clinicopathological characteristics and Tumor Regression Grade.

Variable	No	Total or Partial Tumor Regression 25 Patients (42.3)	Stable or Progressive Disease 34 Patients (57.6)	*p* Value *
Age (years) †				0.5753
<63	32 (54.7)	12 (37.5)	20 (62.5)	
>63	27 (45.3)	13 (48.2)	14 (51.8)	
Gender				0.5190
Male	41 (69.4)	19 (46.3)	22 (53.7)	
Female	18 (30.6)	6 (33.3)	12 (66.4)	
Tumor Site				0.0108 ^‡^
Upper third	25 (42.3)	15 (60.0)	10 (40.0)	
Middle third	8 (13.5)	3 (37.5)	5 (62.5)	
Antrum	18 (30.5)	6 (33.3)	12 (66.4)	
All	8 (13.7)	1 (12.5)	7 (87.5)	
Serum CEA Levels (ng/mL)				0.0519
<3.5	43 (72.8)	22 (51.2)	21 (48.8)	
>3.5	16 (27.2)	3 (18.7)	13 (81.3)	
Serum Ca19-9 Levels (ng/mL)				0.3751
<37	45 (76.3)	21 (46.6)	24 (53.4)	
>37	14 (23.7)	4 (28.5)	10 (71.5)	
Performance Status				0.7334
0	22 (37.3)	12 (54.5)	10 (45.5)	
1	27 (45.8)	7 (25.9)	20 (74.1)	
2	10 (16.9)	6 (60.0)	4 (40.0)	
Tumor Depth				0.0052 ^‡^
T1	2 (3.4)	2 (100.0)	0 (0.0)	
T2	5 (8.4)	3 (60.0)	2 (40.0)	
T3	25 (42.5)	14 (56.0)	11 (44.0)	
T4a	19 (32.2)	4 (21.1)	15 (78.9)	
T4b	8 (13.5)	2 (25.0)	6 (75.0)	
Node Metastasis				0.0373 ^‡^
No	25 (42.3)	15 (60.0)	10 (40.0)	
Yes	34 (57.7)	10 (29.5)	24 (70.5)	
Macroscopic Type				0.0010 ^‡^
1	17 (28.8)	13 (76.5)	4 (23.5)	
2	18 (30.5)	6 (33.4)	12 (66.6)	
3	14 (23.8)	5 (35.7)	9 (64.3)	
4	10 (16.9)	1 (10.0)	9 (90.0)	
Lauren’s Classification				0.0089 ^‡^
Intestinal	25 (42.4)	16 (64.0)	9 (36.0)	
Diffuse/Mixed	34 (57.6)	9 (26.5)	25 (73.5)	
Histological Grade				0.0002 ^‡^
Well Diff.	11 (18.7)	9 (81.8)	2 (18.2)	
Mod. Diff.	8 (13.6)	6 (75.0)	2 (25.0)	
Poor Diff.	40 (67.7)	10 (25.0)	30 (75.0)	
Tumor Size †				0.0001 ^‡^
<3 cm	29 (49.1)	20 (68.9)	9 (31.1)	
>3 cm	30 (50.9)	5 (16.6)	25 (83.4)	
NPS Score				0.0001 ^‡^
0	6 (10.2)	6 (100.0)	0 (0.0)	
1	18 (30.5)	11 (61.2)	7 (38.8)	
2	12 (20.4)	4 (33.4)	8 (66.6)	
3	13 (22.1)	3 (23.0)	10 (77.0)	
4	10 (16.8)	1 (10.0)	9 (90.0)	
NPS Group				0.0001 ^‡^
0	6 (10.2)	6 (100.0)	0 (0.0)	
1	30 (50.9)	15 (50.0)	15 (50.0)	
2	23 (38.9)	4 (17.4)	19 (82.6)	

Numbers in parentheses are percentages. CEA indicates carcinoembryonic antigen (normal value: 3.5 ng/mL); Ca19-9 indicates carbohydrate antigen (normal value: 37 ng/mL); Performance status according to the ECOG scale; * χ^2^ test; ^†^ age and tumor size were dichotomized by median values; ^‡^ significant value.

**Table 2 cancers-13-04676-t002:** Multivariate Analysis with multiple logistic regression.

Variable	Coefficient	Std. Error	r_partial_	t	*p* Value
Age	−0.008855	0.01182	−0.1110	−0.749	0.4575
Gender	−0.2071	0.2628	−0.1167	−0.788	0.4348
Tumor Site	−0.006111	0.1150	−0.007918	−0.0531	0.9579
CEA Levels	0.001770	0.002252	0.1164	0.786	0.4359
Ca19_9 Levels	0.0001714	0.0001764	0.1434	0.972	0.3364
Performance Status	0.1883	0.1955	0.1421	0.963	0.3406
Tumor Depth	0.2336	0.1203	0.2780	1.942	0.0585
Node Metastasis	0.2914	0.2509	0.1706	1.162	0.2515
Macroscopic Type	0.02605	0.1454	0.02670	0.179	0.8586
Lauren’s Type	0.1340	0.05028	0.3692	2.665	0.0107
Histological Grade	0.8161	0.1999	0.5200	4.083	0.0002
Tumor Size	0.6544	0.3450	0.2721	1.897	0.0643
NPS Score	0.2148	0.09443	0.3212	2.275	0.0277
NPS Group	0.4117	0.1807	0.3215	2.278	0.0275

**Table 3 cancers-13-04676-t003:** Performance of different factors to predict neoadjuvant chemotherapy effects.

Variable	AUC	*p* Value	Sensitivity %	Specificity %	+PV %	−PV %
Diffuse Type	0.81 (0.7–0.9)	<0.001	73 (56–87)	80 (59–93)	83 (65–94)	69 (49–85)
Poor Diff.	0.74 (0.6–0.8)	0.001	88 (72–97)	60 (39–79)	75 (59–87)	79 (54–94)
NPS Score 3/4	0.79 (0.6–0.8)	<0.001	79 (62–91)	68 (46–85)	77 (60–89)	71 (49–87)
T3 or T4	0.70 (0.6–0.8)	0.0035	94 (80–99)	20 (7–40)	61 (47–75)	71 (29–96)
Size > 3cm	0.68 (0.5–0.8)	0.0087	73 (56–87)	64 (42–82)	73 (55–87)	64 (42–82)

The performance was computed by using the Receiver Operating Characteristics (ROC) curve analysis; AUC means Area under the ROC curve; +PV means positive predictive value; −PV means negative predictive value; values in parentheses indicate 95% confidence interval; The value for the area under the ROC curve can be interpreted as follows: an area of 0.79 means that the probability of neoadjuvant chemotherapy failure (i.e., TRG > 3) in each patient with a NPS score equal or more than 3 is 79% of the time than that in patient with a NPS score less than 3.

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
