# Peer review of "Naples Prognostic Score Predicts Tumor Regression Grade in Resectable Gastric Cancer Treated with Preoperative Chemotherapy"

_cancers, 2021, doi:10.3390/cancers13184676_

Round 1

Reviewer 1 Report

The present article is a continue a line of research conducted by the main authors. Indeed, after the study of Naples Prognostic Score in patients undergoing surgery for colorectal cancer and the rule of the inflammatory and nutritional status in patients undergoing surgery for gastric cancer, the authors decided to investigate the impact of the Naples Prognostic Score as predictor for the tumor regression grade in resectable gastric cancer treated with preoperative chemotherapy.

The article has a simple structure and it could represent an interesting precondition to test the utility of the Naples Prognostic Score as predictor also in this kind of population. However, since the study design is a retrospective analysis of a consecutive case series of 59 patients and the context being monocentric, the internal and the external validity of the results will be inevitably limited by these points.

My main and major concerns are about statistical analysis.

-Could you please further detail how you tested the difference in distribution among patients with total/partial response against none/worst group in case of variables that have more than 2 modalities? The chi square test does not always appear to be the most appropriate test also for the presence of 0 or low frequencies in the contingency tables. Also in table 1, it could be helpful to add percentages.

-One of my main worries concerns multivariable logistic regression. Considering the sample size, the number of regressors simultaneously considered could be problematic and afflicted by multicollinearity. A stepwise approach is recommended. Keep particular attention to the simultaneous inclusion of variants of NPS score in the same model. Since they are substantially the same variable, I suggest performing a sensitivity analysis by implementing two independent models and, if the results are approximately the same, to show the one that includes only the variable with 3 levels.

-My second point concerns the analysis of the ROC and AUC curves. I am not convinced by the robustness of the results, and I am skeptical because the random chance generated by the small sample size further divided by the multiple modalities. Could you bootstrap or cross-validate the results with an appropriate statistical method?

-Could you provide a distribution of the 59 patients across the study period divided by tumor regression grade?

I strongly suggest including a qualified Statistician in your work.

Author Response

Reviewer #1: The present article is a continue a line of research conducted by the main authors. Indeed, after the study of Naples Prognostic Score in patients undergoing surgery for colorectal cancer and the rule of the inflammatory and nutritional status in patients undergoing surgery for gastric cancer, the authors decided to investigate the impact of the Naples Prognostic Score as predictor for the tumor regression grade in resectable gastric cancer treated with preoperative chemotherapy.

The article has a simple structure and it could represent an interesting precondition to test the utility of the Naples Prognostic Score as predictor also in this kind of population. However, since the study design is a retrospective analysis of a consecutive case series of 59 patients and the context being monocentric, the internal and the external validity of the results will be inevitably limited by these points.

Reply: We thank the reviewer for the correct analysis of our work. We fully agree with his/her consideration that our results need further validation. By publishing it in Cancers we hope to stimulate a such study.

-Could you please further detail how you tested the difference in distribution among patients with total/partial response against none/worst group in case of variables that have more than 2 modalities? The chi square test does not always appear to be the most appropriate test also for the presence of 0 or low frequencies in the contingency tables. Also in table 1, it could be helpful to add percentages.

Reply: the comment of the Reviewer is very argue. In these instances the statistical software performs the Chi-square test for trend. The Cochran-Armitage test for trend (Armitage, 1955) tests whether there is a linear trend between row (or column) number and the fraction of subjects in the left column (or top row). The Cochran-Armitage test for trend provides a more powerful test than the unordered independence test.  In addition, we added percentages to the Table 1.

-One of my main worries concerns multivariable logistic regression. Considering the sample size, the number of regressors simultaneously considered could be problematic and afflicted by multicollinearity. A stepwise approach is recommended. Keep particular attention to the simultaneous inclusion of variants of NPS score in the same model. Since they are substantially the same variable, I suggest performing a sensitivity analysis by implementing two independent models and, if the results are approximately the same, to show the one that includes only the variable with 3 levels.

Reply: We are grateful to the Reviewer raising this important issue. As stated above, our data are always analyzed by a skilled biostatistician, who has also taken part in revision of this manuscript. In statistical analyses, multicollinearity is an event in which two or more variables in the multivariable logistic regression are highly correlated, and one can be linearly predicted from the others, and it affects calculations regarding individual predictors. To investigate multicollinearity the SPSS software allows to use analysis with interaction that, joining together the variables, evaluate their coefficient and p value. If p value is significant, it means that the two variables correlated and model could be incorrect (Chan YH. Biostatistics 203. Survival Analysis. Singapore Med J 2004;45:249-256). We were fully aware of this issue, particularly for the following covariates: parameters T and N, Lauren's type and histological grade, and NPS score and NPS group, as correctly pointed out by the Reviewer. Thus, the multivariate analysis with multiple logistic regression we presented in the Table 2 should contain also interaction analyses and this would make substantial complexity of the Table 2 that could be incomprehensible to the reader. For this reason we did not present it in this fashion. However, we agree with the Reviewer that this issue should be clarified in the manuscript. We added the following sentences ' Multicollinearity among variables supposed to have high correlation was investigated with interaction analysis. ' (see paragraph Methods, subparagraph Statistical Analysis, of this revised manuscript); and ' where interaction analysis excluded any correlation.' (see paragraph Results, subparagraph Comparison of the Predictive Capability of the Staging Systems).

We regret for our mistake and forgetfulness in the last rows of the Tables 1 and 2 lacking of the term group for NPS. As wittily and correctly supposed by the Reviewer, since NPS score and NPS group are substantially the same variable, we analyzed two different models for each method used to show NPS values. In both models the statistical results were the same (as expected due to absence of multicollinearity shown by interaction analysis).

Ultimately, in this study we hope to have correctly shown that NPS correlate with tumor regression grade regardless of using NPS score or NPS group. In addition, we have added the term group to the NPS, both in the Table 1 and 2.

-My second point concerns the analysis of the ROC and AUC curves. I am not convinced by the robustness of the results, and I am skeptical because the random chance generated by the small sample size further divided by the multiple modalities. Could you bootstrap or cross-validate the results with an appropriate statistical method?

Reply: Once again the Reviewer is very right and his/her comment particularly insightful. In truth, the ROC curve analyses were performed by using the MedCalc software and it has been now stated in this amended manuscript (see paragraph Materials and Methods, subparagraph Statistical Analysis, of this manuscript). In all cases, the results were obtained computing 1000 bootstrap replications, the bias-corrected and accelerated (BCa) bootstrap adjusted for possible bias and skewness in the bootstrap distribution, and the estimation of sensitivity and specificity at fixed specificity and sensitivity together with bootstrap Youden index confidence interval identified the optimal criterion for each prognostic variable. These were NPS 3/4, diffuse type, and poor differentiation.

-Could you provide a distribution of the 59 patients across the study period divided by tumor regression grade?

Reply: All the 59 patients were treated by the same multidisciplinary oncological team. Particularly neoadjuvant chemotherapy and surgical strategy were identical. However, we too have asked ourselves the same question, and we explored this topic. In the first 20 patients TRG was 1-3 in 9 and 4-5 in 11 patients, respectively. In the following 20 patients, the TRG was 1-3 in 8 and 4-5 in 12 patients, respectively. In the remaining 19 patients, TRG was 1-3 in 8 and 4-5 in 11 patients, respectively. Thus, non differences were seen in tumor regression grade across different periods of the study.

We hope to have fulfilled the comments and suggestions of the Reviewer whom once again we thank very much.

Reviewer 2 Report

The authors examined gastric cancer with preoperative chemotherapy to validate utility of Naples Prognostic Score (NPS), and NPS is correlated with tumor regression grade (TRG).  As NPS represents immune-nutrition conditions in the preoperative clinics, it can be helpful to tailored made therapy in gastric cancer.  There are several concerns for publication.

1, Results should be structured in each paragraph.

2, Please clarify the difference between NPS score and NPS in Table 1.

3, Please show prognostic curve by NPS score and NPS in Fig. 2.

4, Definition of NPS should be explained before the initial use in the result session.

5, I think that NPS and NPS score are confounding, and that Lauren histology and histological differentiation are confounding, either, and it is not appropriate in the multivariate analysis.

6, References 15 and 16 are not related to-FLOT-based chemotherapy in discussion.

7, I can not agree with rationale of NPS score.  I ask you more suitable and scientific  explanation that it can reflect immune-inflammation status.

Author Response

"please see the attachment"

Reviewer #2:

The authors examined gastric cancer with preoperative chemotherapy to validate utility of Naples Prognostic Score (NPS), and NPS is correlated with tumor regression grade (TRG).  As NPS represents immune-nutrition conditions in the preoperative clinics, it can be helpful to tailored made therapy in gastric cancer.  

Reply: We thank very much the Reviewer who well summarized our study.

There are several concerns for publication.

  1. Results should be structured in each paragraph.

Reply: According to the Reviewer's suggestion, The paragraph Results has been structured in the following three subparagraphs:

- Clinical Characteristics

- Tumor Regression Grade and its Prognostic Predictors

- Tumor Regression Grade, Surgical Radicality and Survival Rate

  1. Please clarify the difference between NPS score and NPS in Table 1.

Reply: As outlined in the reply to the Reviewer #1, we regret for our mistake and forgetfulness in the last rows of Tables 1 and 2 lacking of the term group for NPS (now we have added it in this revised manuscript). However, NPS score ranges from 0 to 4 (assuming from each variable, included in the score, a value of 0=normal, and 1=pathological, NPS score is the sum of each factor included in the model); on the other hand, NPS group ranges from 0 to 2 (group=0 means no alterations; group=1 means one or two pathological values, group=2 means three or four pathological alterations).

  1. Please show prognostic curve by NPS score and NPS in Fig. 2.

Reply: The aim of this study was to identify possible predictors of tumor regression grade in gastric cancer patients undergoing neoadjuvant chemotherapy. The Figure 2 in the submitted manuscript was needed to clarify that TRG is correlated with long-term outcome [in all (Figure 2A), and radically resected (Figure 2B) patients, respectively). Significant correlations between NPS and overall survival rate has been already demonstrated in several human tumors, including our experience in gastric cancer patients (ref. n° 33 of this amended manuscript), and probably is outside of the aim of this study.

If necessary, we are ready to submit this survival curve that we added below:

  1. Definition of NPS should be explained before the initial use in the result session.

Reply: In the paragraph Introduction the four constituent factors of NPS are listed. In addition, in the paragraph Materials and Methods, subparagraph Naples Prognostic Score, is stated how to calculate the score. If necessary, according to the Reviewer's suggestion we could add further details.

  1. I think that NPS and NPS score are confounding. and that Lauren histology and histological differentiation are confounding. either. and it is not appropriate in the multivariate analysis.

Reply: The comment of the Reviewer is correct and very insightful. We are grateful to the Reviewer raising this important issue, and we can reassure that our data are always analyzed by a skilled biostatistician, who has also taken part in revision of this manuscript. In statistical analyses, multicollinearity is an event in which two or more variables in the multivariable logistic regression are highly correlated, and one can be linearly predicted from the others, and it affects calculations regarding individual predictors. To investigate multicollinearity the SPSS software allows to use analysis with interaction that, joining together the variables, evaluate their coefficient and p value. If p value is significant, it means that the two variables correlated and model could be incorrect (Chan YH. Biostatistics 203. Survival Analysis. Singapore Med J 2004;45:249-256). We were fully aware of this issue, particularly for the following covariates: parameters T and N, Lauren's type and histological grade, and NPS score and NPS group, as correctly pointed out by the Reviewer. Thus, the multivariate analysis with multiple logistic regression we presented in the Table 2 should contain also interaction analyses and this would make substantial complexity of the Table 2 that could be incomprehensible to the reader. For this reason we did not present it in this fashion. However, we agree with the Reviewer that this issue should be clarified in the manuscript. We added the following sentences ' Multicollinearity among variables supposed to have high correlation was investigated with interaction analysis. ' (see paragraph Methods, subparagraph Statistical Analysis, of this revised manuscript); and ' where interaction analysis excluded any correlation.' (see paragraph Results, subparagraph Comparison of the Predictive Capability of the Staging Systems, of this manuscript).

  1. References 15 and 16 are not related to-FLOT-based chemotherapy in discussion.

Reply: The Reviewer's observation is fully right. In the paragraph Introduction we used such references to state that some factors have been demonstrated to correlate with chemotherapy efficacy, particularly in preoperative setting. The statement was replicated in the paragraph Discussion where instead it was related to FLOT therapy. Thank you very much; we have changed the bibliographic reference.

  1. I can not agree with rationale of NPS score.  I ask you more suitable and scientific  explanation that it can reflect immune-inflammation status.

Reply: The Reviewer's consideration is very shrewd. However, the scientific basis of NPS have been widely elucidated in several previous studies reports dealing with this topic. Particularly in the articles cited in our current manuscript (please see references n° 28, 32, 33, 50-56) accurate and extensive explanations of the immune, nutritional and inflammatory mechanisms and of the correlations with NPS components have been made. For example, the following statements were reported in our previous article to explain importance of albumin and cholesterol levels:

' Particularly,  hypoalbuminemia is  a marker not only of malnutrition but also of systemic inflammation since some pro-inflammatory substances, such as cytokines, reduce the concentration of albumin.15 Serum albumin levels are present in all scoring systems.9,10,12,14 However, since albumin concentrations can be affected by liver function and changes in body fluid volume,15 some authors have proposed adding plasma cholesterol levels to optimize the evaluation of nutritional status.5,10 Low cholesterol levels have been reported to correlate with poorer prognosis in many human tumors, including CRC.24,25 Hypocholesterolemia influences cell membrane fluidity, decreasing the mobility of cell surface receptors and their ability to transmit transmembrane signals.26 Consequently, immunocompetent cells become unable to destroy cancer cells with changes in their membranes.10,27 etc.'

In order to avoid unnecessary repetition and reduce the length of the article, we preferred give just a few hints related to NPS mirroring immune-inflammation status.

We believe that the reader of this article can access the cited bibliography as well as the previous one and get an idea of the importance of the NPS.

We hope to have fulfilled the comments and considerations of the Reviewer whom once again we thank very much. 

Round 2

Reviewer 1 Report

The authors have provided a depth justifications and plausible explanations. In relation to the nature of the study and the relevance of the results, eventual statistical limits could be considered as not determinant to acceptability the manuscript.

Author Response

To Reviewer #1

Thank you very much for your favourable comment.

Reviewer 2 Report

Thank  you for your revisions.  I am satisfied with all revision but comment 3.  KM curve is required for publication, because some readers would like to know its information for clinical reference.

Author Response

To Reviewer #2

Thank you for suggestion. The KM figure 3 has been added to the amended manuscipt.